# Addressing information and credit barriers to making India open defecation free and improving child health: Evidence from a cluster randomized trial in rural India

**Payal Seth**[ID]*, **Prabhu Pingali**

Tata-Cornell Institute for Agriculture and Nutrition, Cornell University, Ithaca, New York, United States of America

* payalseth1309@gmail.com

## Abstract

### Background

Open defecation (OD) remains a significant public health challenge in India, contributing to adverse child health outcomes. Eliminating OD and improving child health necessitates both *universal* access and adoption of toilets. Despite the success of removing credit constraints and enhancing access to subsidized toilets through national sanitation campaigns, the adoption of these toilets is still lagging in India. This is because households might also be lacking information about the benefits of using toilets (information constraint).

### Methods

In this paper, we test for the joint efficacy of the removal of information and credit constraints versus solely addressing the credit constraint at eliminating OD, based on a cluster randomized control trial in rural India. We implemented two interventions: a *universal* community-led behavior change campaign along with subsidized construction of individual household toilets for *every* household that opted for subsidy (cluster A) and only subsidized construction of individual household toilet construction for *every* household that opted for the subsidy (cluster B). No behavior change was provided in cluster B. The control group did not receive any intervention.

### Results

We find that the removal of information and credit constraints at a *near-universal* level in cluster A resulted in improved toilet access and adoption, eliminating OD at the community level, with a significant gain in child weight-for-age z-scores (WAZ scores). There was an increase in the percentage of households that owned individual household toilets from 2% to 98% and a decline in female respondents practicing OD from 98% to 4% in cluster A. In cluster B, ownership of individual household toilets improved by 78 percentage points (from 6% to 84%), but OD decreased by only 45 percentage points (from 95% to 50%) for

**Data availability statement:** The data can be found at https://osf.io/hmkdw/

**Funding:** The study was funded by Bill and Melinda Gates Foundation (# OPP1137807). The funders had no role in study design, data collection and analysis, decision to publish, or preparation of the manuscript.

**Competing interests:** The authors have declared that no competing interests exist.

the respondents. The control group saw no significant changes. For children under five, there was a statistically significant increase in WAZ scores by 0.68-0.69 standard deviations in cluster A, while cluster B showed insignificant changes when compared to the control group.

## Conclusions

The study implies that supplementing *universal* financial support with *community-level* information intervention enhances sustainable adoption of improved sanitation facilities, aiding India's progress towards an open defecation-free nation and improving child health outcomes.

## 1. Introduction

United Nation's Sustainable Development Goal (SDG) 6 calls for ensuring clean water and sanitation for all by 2030. Improving access to sanitation facilities has been a public sector priority in India as historically, India has been home to the largest number of people practicing open defecation (OD) in the world [1]. Although the national sanitation campaigns have been successful in enhancing access to toilets, yet 17% of the rural population is still practicing OD in India [2]. Poor sanitation defined as improper disposal of human waste can contaminate water sources, food, and surfaces and increase the risk of diarrhoeal disease and other infections which has grave consequences on the short and long-term impacts on human development especially among the vulnerable demography of children below five years of age [3,4]. Consequently, India is also host to the world's highest number of children with severe acute malnutrition [5]. Hence, improving sanitation conditions is often envisioned as a critical strategy to reduce the burden of child malnutrition. In this paper, we report the results from a cluster randomized controlled trial (CRCT) of a sanitation intervention that was successful at combatting OD and improving child weight outcome in rural India.

The key to making India open defecation free (ODF) and improving child health outcomes is securing *universal access* as well as the *adoption* of toilets. While *access* to toilets is hindered by credit constraint, there is an information constraint as well that can impede the successful *adoption* of toilets.

Credit constraints refer to the lack of access to credit, which can prevent households from building toilets, hence its removal has been identified as a significant factor in lowering the cost of OD, making it convenient for people to use toilets [6,7]. The previous national sanitation campaigns, like the Total Sanitation Campaign (1999-2012) and the latest Swachh Bharath Mission (translated as Clean India Mission) (2014-2019) have dealt with this issue largely from a supply-side perspective by removing credit constraints through subsiding toilet construction, with limited implementation of behavior change component on the field [8–10].

Information constraint refers to the lack of knowledge about the importance of sanitation, the risks associated with OD, and the benefits of using toilets. Research suggests that credit constraints alone do not explain the persistence of OD in rural India. An equally critical role is played by the removal of information constraints that lead to demand-induced adoption of toilets [8,10–14]. This is because OD is a deeply-rooted, centuries-old social norm in India, which exists not only due to poverty but also due to the lack of awareness of the benefits of using toilets [13]. Disseminating sanitation-related behavior change communication (BCC) campaign is expected to raise awareness regarding the adverse consequences of OD, generate demand for toilets, and lead to an increase in the use of toilets.

A host of international and national experimental studies on the sanitation-child health nexus have reported a less-than-universal increase in *access* to toilets within the treated communities, no or little increase in toilet *adoption*, and consequently have failed to translate to any significant improvements in child health [10,12,15–25].

It is in this context that the literature highlighted the urgency to undertake experiments that identify the precise nature of sanitation interventions (a combination of removal of credit and/or information constraints) that would prove to be most effective at securing both universal access and adoption, i.e., elimination of OD [26,27].

In this paper, we address this research gap by analyzing the effects of the removal of both credit and information constraints versus the removal of just credit constraint, from a CRCT conducted in 15 villages of Uttar Pradesh (U.P), India. In the first arm of five villages (cluster A), we removed both the information and credit constraint by providing a *community-wide* BCC sanitation campaign prior to building subsidized individual household toilets for *every* household that opted for the subsidy. In the second intervention arm (next five villages, cluster B), we only removed the credit constraint, by building subsidized individual household toilets for *every* household that chose to utilize the subsidy, without the BCC component. The remaining five villages served as the control villages.

Using this research design, we answer these two important research questions. First, d*oes the removal of both information and credit constraints (versus just the credit constraint) lead to higher adoption of* toilets? Second, since the adoption of toilets is likely to contain the spread of fecal matter, thereby eliminating infections: *Does this higher adoption of toilets also lead to statistically significant gains in child health?*

The results from the CRCT inform us that the almost *every* household in cluster A villages agreed to build and use individual toilets. This resulted in a *near-universal* increase in *access* as well as *adoption* of sanitation facilities. Furthermore, the transition from *near-universal* OD to the adoption of toilets led to a significant increase of about 0.68-0.69 standard deviation of weight-for-age z-score in children below five. In cluster B villages, *access* to toilets increased but this *access was not universal*. Hence, the *adoption* rates were also *not universal* and we found *no statistically significant improvements in child health*.

Our experimental results indicate that the households suffered from both information and liquidity constraints. Learnings from this experiment imply that sanitation campaigns ought to incorporate behavior change campaigns along with interventions that help improve household toilet access.

The paper is organized as follows. Section 2 provides a contextual background on the barriers to toilet adoption in India and mechanisms that link improved sanitation with child health. Section 3 discusses the research design, its contribution, and the timeline of the study. Section 4 explains the data collection methodology, summary statistics, empirical strategy, and the outcomes used in the study. Section 5 presents the results or the impact of two interventions on the study outcomes. Finally, section 6 discusses the contributions and concludes with policy implications.

## 2. Section 2: Background

### 2.1. Understanding barriers to toilet adoption in India

**2.1.1. Limited success of toilet adoption and impact on child health during national sanitation campaigns.** In this sub-section we highlight how the limited success of toilet adoption and its impact on child health during national sanitation campaigns could be attributed to (a) inadequate increase in toilet coverage, (b) inadequate toilet quality, and (c) low emphasis on behavioral change.

To combat the enduringly stubborn challenge of OD in India, the government of India launched its national sanitation campaign, called the Total Sanitation Campaign (TSC) to accelerate toilet coverage in India in 1999. TSC partially subsidized hardware construction for below poverty line households but implementation and effectiveness of behavioral change components were limited [7,9,10,28]. Four previous studies used cluster-randomized trials to evaluate the impact of TSC on toilet coverage, use, and child health outcomes. Pattanayak et al. 2009 found a modest increase of 19 percentage points in toilet ownership using difference-difference estimates in 20 treatment villages after 5-7 months of implementation of the TSC program in 2006 in two blocks of the state of Orissa. A study conducted in 80 villages in Madya Pradesh reported a modest increase of 19 percentage points in toilet ownership and a decrease of 10 percentage points in open defecation in the treatment villages after two years of the program [10]. The decline was not substantive enough to translate into any improvements in child health. Hammer and Spears 2016 led the evaluation of TSC in one district of Maharashtra over 18 months in 2004. The program led to a modest 8.2 percentage point increase in toilet ownership in treatment villages and was associated with a 0.3-0.4 standard deviation increase in children's height-for-age z-scores. TSC was also evaluated by Clasen et al. 2014 in 100 villages in Odisha from 2010 to 2013 leading to an increase of 54 percentage points in toilet ownership in treatment villages, a modest increase of 36% in toilet use, and no impact on child health. It is important to note that even a small fraction of people open defecating exposes the whole community to the risk of transmission of fecal pathogens, implying that inadequate sanitation imposes negative externalities and the collective level of sanitation coverage and usage in a village is an important determinant of health outcomes [29–32]. Therefore, the limited success of TSC can be attributed to the inadequate increase in toilet coverage and consequently low increase in toilet use to reach that minimum necessary threshold to withhold the transmission of fecal pathogens and lead to improvement in health outcomes [10].

However, even when high toilet coverage was achieved across the communities, toilet use remained low. Barnard et al. 2013 conducted a cross-sectional analysis of toilet construction under TSC and its use in 20 villages in Orissa, India. The authors found that although the program made a substantial gain in coverage (about 72% in treatment villages), about 40% of them were not being used by any member of the household. A study from the five northern states of India concluded that 60% of the households that received toilet construction materials from the government had at least one member practicing OD [13]. This is because the mean pit size of government-constructed toilets was nearly half of the size of the privately constructed toilets and incorrect perceptions about the frequency of emptying them impeded the adoption. Besides technical limitations, behavioral challenges remain to be resolved too. The authors also state that 47% of people who defecated in the open explained that they did so because there was a preference for OD even in the presence of toilets.

**2.1.2. Low toilet adoption even in the presence of behavioral change campaigns under SBM.** TSC was later renamed Nirmal Bharat Abhiyan and further restructured as Swachh Bharath Mission (SBM) (translated as Clean India Mission) in 2014 which aims to make India open defecation free by 2019. The mission's primary goal was to eliminate the practice of open defecation by 2019 by constructing household toilets and promoting behavioral change among the population [33].

While there was an emphasis on behavior change in the program, evidence suggests that coercive tactics played a more prominent role than genuine behavioral shifts. In a study covering Uttar Pradesh, Rajasthan, Bihar, and Madhya Pradesh, Gupta et al. 2019 reported that 44% of rural people over two years of age defecate in the open, and a quarter of people who owned a government-constructed toilet still preferred OD. The authors report that more than half of the households had heard of coercive tactics being employed to achieve construction

completion and use. The coercive measures employed to increase latrine usage, particularly among Scheduled Tribe and Scheduled Caste households, raise concerns regarding the genuinity and sustainability of the behavioral changes achieved. The authors conclude that to successfully eliminate OD, intimidation tactics should end and instead social attitudes towards OD should be targeted to encourage sustained toilet use.

Similarly, a more recent cluster-randomized impact evaluation in rural Punjab revealed that the impact of SBM intervention was modest in reducing OD. While the behavior change campaigns had a significant positive impact on improving awareness of handwashing and hygiene-related behavior among school-going children, the campaign did not reach a large majority of the beneficiary households [34].

**2.1.3. Giving ownership to enhance toilet adoption.** Solely providing subsidies without behavior change, suffers from the critique of undervaluing intrinsic motivation in adopting a healthy behavior [35]. Coffey et al. 2014 find that households that chose to build their own latrine were more likely to use one. Gupta et al. 2019 found that households that received money to build their own toilet rather than have the government construct one for them, were almost 10 percentage points less likely to defecate in the open. Gertler et al. 2015 go further ahead and state that the installation of private sanitation will have a much larger impact on the reduction in OD than any behavior change intervention.

## 2.2. Mechanism linking improved sanitation and child health

This sub-section highlights the literature from biomedical sciences to understand the pathways through which water, sanitation, and hygiene interventions can lead to improved health. Three biological mechanisms link inadequate sanitation with undernutrition: diarrhea, intestinal infections, and environmental enteropathy [36]. Diarrhea is the result of infections from consuming food or water contaminated with feces, or even direct ingestion of fecal matter [37]. Soil-transmitted helminth infections are caused by soil contaminated with eggs present in human feces. These eggs are either ingested from nearby contaminated water sources or children's soiled unwashed fingers after playing in the soil [38]. Repeated exposure to infections changes the structure of the gut, leading to impaired immunity and stunting without overtly manifesting into diarrhea [36]. However, the adverse consequences from the fecal-oral route can be prevented easily as the transmission occurs in the domestic and community environment of the child. It is in this context that hygiene and sanitation interventions play an important role in reducing the occurrence of diseases.

The F-diagram by *Curtis, Cairncross, and Yonli, 2000* (see Fig 1) sketches all the potential pathways of transmission of fecal matter to the host [39]. It is also helpful to visualize the impact of interventions introduced at different stages. Primary barriers are employed to prevent the entry of feces into the environment, while secondary barriers stop the pathogens that have already entered the environment from infecting the host [39]. Hygiene promotion campaigns, in addition to promoting toilet construction, can help prevent environmental contamination and disease spread.

# 3. Section 3: Research design, its contribution, and timeline

## 3.1. Research design and its contribution

Understanding the barriers to access and adoption of toilets in India has led us to a research design that incorporates all the limitations identified in previous studies:

1. **Ensuring Access to High-Quality Individual Household Toilets (Removal of Credit Constraints):** Past initiatives have struggled with inadequate coverage and substandard

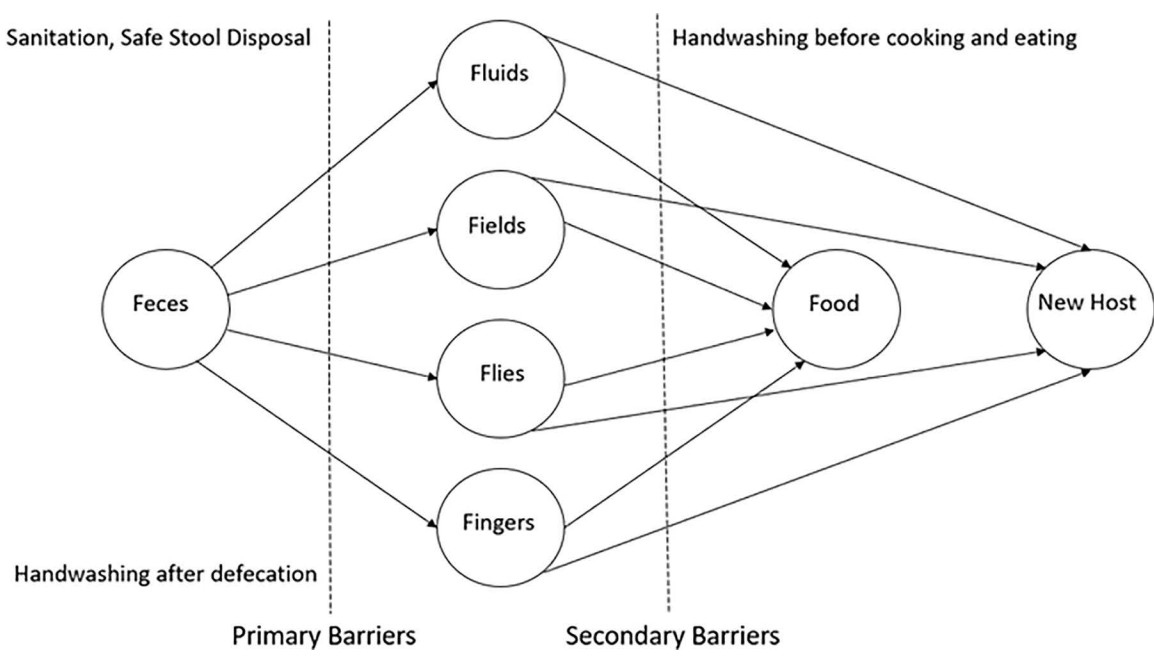

**Fig 1. From Fig 2 of Curtis, Cairncross, and Yonli, 2000.**

toilet construction, often leading to a high proportion of dysfunctional facilities. This research design emphasizes access to high-quality individual toilets for *every* household.

2. **Fostering Toilet Ownership and Investment:** Evidence suggests that when households are involved in the construction of their own toilets, they are more likely to adopt and use them consistently. Our approach aims to provide financial and in-kind contributions that promote a sense of ownership, thereby increasing the likelihood of sustained toilet use.

3. **Implementing** *Universal***, Rigorous Behavioral Change Communication (BCC) Campaigns (Eliminating Information Constraint):** Previous campaigns have shown that behavioral change initiatives alone may not be sufficient to eliminate OD. Our research design will integrate a robust BCC strategy that not only engages *all* the members in the communities but also promotes a sustained cultural shift towards improved sanitation practices.

In partnership with the local non-governmental organization, Grameen Development Services (GDS), we implemented a cluster-randomized control trial (CRCT) in the Maharajganj district of eastern Uttar Pradesh, India, involving 15 villages and 866 households. This district was chosen as our 15 study villages served as part of Tata-Cornell Institute (TCI), Cornell University's Technical Assistance and Research in Indian Nutrition and Agriculture (TARINA) (S. Gupta et al., 2019). Maharjganj district was also well suited to our study as according to NFHS-4 (2015-16) this district had only 17% toilet coverage in the rural areas and low exposure to any sanitation program. Further, the district head agreed to not undertake any sanitation intervention in the control villages until the completion of the experiment.

The randomization of this CRCT study design was done at the village level with an equal allocation of villages to the two treatment arms and one control arm. Five villages randomized to the cluster A arm received the community-level BCC campaign that uses the tenets of

community-led total sanitation (CLTS), along with the construction of subsidized individual household toilets for *every* household that chooses to avail the subsidy. The five villages randomized to cluster B group were provided with the same choice of building toilets as was given to cluster A but without any BCC intervention. The five villages randomly assigned to the control group did not receive any intervention but participated in the survey rounds. All 866 households across the 15 villages participated in the study, representing a comprehensive census of the entire population. The study protocol was approved by the Institutional Review Board at Cornell University and informed consent from the respondents as well as the primary caretaker of the child was recorded electronically (Protocol number: 1609006597). The treatment arms are described in detail below.

**Cluster A.** In Cluster A of our study, all households received the **Community-Led Total Sanitation (CLTS)** campaign. CLTS was developed by Kamal Kar, a development consultant from Bangladesh in 2000 and is now implemented in more than 60 countries across the world [40]. CLTS is the practice of influencing sanitation behavior at the community level by highlighting the adverse consequences of OD and using a series of public exercises on how to become an ODF community. CLTS was implemented in the field to empower communities to take responsibility for their sanitation and hygiene practices. The intervention began with "triggering" sessions, where trained facilitators led the community through activities designed to raise awareness about the dangers of open defecation and its impacts on health. These sessions included mapping of open defecation areas, role-playing exercises, and visual demonstrations to help community members understand the environmental and health risks associated with poor sanitation. The triggering process aimed to inspire a collective sense of responsibility and motivate the community to take action. Following the triggering sessions, community members were encouraged to participate in action planning to identify local solutions for improving sanitation. This campaign was conducted in the presence of all households in the village to ensure widespread participation and engagement, **eliminating information constraint.**

After the dissemination of the BCC campaign, **temporary (kuchha) toilets** were built for *all* households in Cluster A. These toilets were used for 4-5 months, providing an immediate solution to sanitation needs while the community worked toward more permanent solutions. The temporary toilets were an essential short-term measure to meet demands from the CLTS campaign to reduce open defecation and improve hygiene. BCC sessions also imparted knowledge regarding the importance and correct technique for hand washing after defecation as well as stool disposal. Finally, **individual household toilets** were built for *every* household that chose to avail of the subsidy offered by the intervention. Under this scheme, TCI covered 75% (Rs 12000, USD 180- an average exchange rate of Rs 67 per dollar in 2016 was used to calculate this figure) of the construction cost, with households contributing the remaining 25%, either in cash or in-kind. This offered a significant subsidy for households that wanted to invest in permanent sanitation infrastructure but needed financial support to do so, **thereby removing the credit constraint**.

**Cluster B.** Cluster B, the intervention focused on providing **individual toilets** for households that opted to avail themselves of the subsidy. TCI built individual toilets for these households, covering 75% of the construction cost, while households were responsible for contributing the remaining 25%, either in cash or in-kind. Unlike Cluster A, Cluster B did not receive the temporary kuccha toilets or the community-level behavior change campaign based on CLTS. Instead, the focus was solely on providing long-term sanitation solutions through subsidized toilet construction for households that expressed interest in building their own toilets. This approach aimed to improve sanitation access directly by **relieving** the households of the **credit constraint.**

**Control group.** The control group comprised households that received no interventions. They participated solely in survey rounds, providing a baseline against which the effects of the interventions in Clusters A and B could be measured.

Both treatments of BCC campaigns and building of subsidized individual household toilets were assigned at the village level and given to *all* households. This is a common practice in the sanitation-nutrition literature as OD emanates negative externalities. That is, as long as some fraction of members in the community are practicing OD, their neighbors, even if using toilets, are still exposed to fecal pathogens and prone to infection [41]. Hence, to ensure that the elimination of OD happens at the community level, the interventions are targeted towards all the households in the village.

See Table 1 for a summarised overview of the interventions.

Our research design is unique and contributes to the literature in several ways. First, our study recognizes the externalities imposed by OD as even the members of the community who do not practice OD still live in an environment exposed to fecal pathogens. Therefore, our program provides the option to build toilets for *every* household in the 10 treatment villages and by providing the necessary technical support to build good quality toilets (removal of credit constraints). Second, this design mandates that households bear the rest of the 25% of the cost giving them a stake in ownership and making them more likely to use the toilet [8,14,42,43]. Third, our research design acknowledges the widespread preference for open defecation and consequent complementarities between sanitation campaigns and toilet adoption. Hence, half of the treatment villages receive community-wide behavior change intervention about the issues of OD before the option of construction is presented to them (elimination of information constraint). Therefore, our treatment in cluster A imparts both *universal* access to toilets and information. Our study recognizes that ensuring high sanitation coverage and use as well as the provision of information to *all* the participants is the fundamental foundation of an effective sanitation strategy.

## 3.2. Experiment timeline

The female head of the household was our primary respondent in the survey. The study was conducted between 2016 and 2019 and its timeline is depicted in Fig 2.

Prior to the intervention, we administered a baseline survey for all 866 households in the 15 villages in June-July 2017. It included modules on socioeconomic characteristics of the household, hygiene practices, defecation behavior, and preferences, among other questions. In

**Table 1. Details of intervention by groups.**

| Group | Intervention type | Number of villages |
|---|---|---|
| **Cluster A** | 1. Community-level behavior change campaign based on community-led total sanitation (CLTS), conducted in the presence of all household members in the village.<br>2. Temporary (kuccha) toilets built for all households, used for 4-5 months to address immediate sanitation needs.<br>3. Building individual household toilets: TCI built individual toilets for *every* household that opted for the subsidy. TCI covered 75% of the cost while the households contributed the remaining 25% of the cost (either in cash or in-kind). | 5 |
| **Cluster B** | Building individual household toilets: TCI built individual toilets for households that opted for the subsidy. TCI covered 75% of the cost while the households contributed the remaining 25% of the cost (either in cash or in-kind).<br>No community-level BCC campaign or temporary toilets | 5 |
| **Control Group** | No intervention provided; participated in survey rounds | 5 |

| 2016 | August- September | Sample Selection and Randomization of 15 villages for the study |
|---|---|---|
| 2017 | June- July | Baseline Data Collection<br>Behavior change intervention was completed by July 31, 2017 |
| | August- December | Kucha toilets built and used by the households |
| 2018 | January | Construction of toilets was completed by January 31, 2018 |
| | February- December | Monthly Follow-up Toilet Use Survey |
| | December | Endline Data Collection |
| 2019 | January | Endline Data Collection |

**Fig 2. Timeline of the intervention and data collection exercise.**

addition, we also collected information on children's nutrition outcomes as captured through anthropometric measurements for all 278 children below five years of age in the 15 villages. The BCC implementation exercise was completed by the end of July 2017 and toilet construction was finalized by January 31, 2018.

From February 2018 to December 2018, we conducted monthly surveys with households to track toilet adoption trends. These short follow-up surveys involved interviewing both the household heads and the primary female respondents about their toilet use. Additionally, enumerators verified toilet usage during these visits. This approach allowed us to capture variations in toilet adoption across different interventions, genders, and seasonal patterns, providing a comprehensive understanding of the trends over time. We did not seek information on other modules from the baseline survey in these monthly surveys.

The final endline survey was carried out in December 2018 and January 2019, 18 months after the behavior change dissemination and construction of temporary toilets, and around 11-12 months after the completion of toilet construction. The modules were similar to the baseline survey.

Of the 866 households covered in the baseline survey, 846 (97.6 percent) participated in the endline survey. Of the 846 households, 339 (40%) belonged to Cluster A, 270 (32%) belonged to Cluster B, and 237 (28%) in the control arm. Of the 278 children who were surveyed in the baseline, only 213 children who were less than five years of age in the baseline and endline were included for analysis. A panel dataset of the same set of 846 households in the baseline and endline and a panel of 213 children in the baseline and endline are used for the empirical analysis.

## 4. Section 4: Data description, empirical strategy and outcomes

### 4.1. Descriptive statistics and balancing test

Table 2 presents the summary statistics of the sanitation profile of the respondents and the households as well as child characteristics from the baseline survey. More than 95% of the respondents were practicing OD and only 5% of the households had a toilet in the baseline. Next, even though around 40% of the households had soap near the handwashing station, only

**Table 2. Baseline summary statistics on the sanitation profile of the households and respondents.**

| | Sample mean | Observations |
|---|---|---|
| | (1) | (2) |
| Number of villages | 15 | |
| Number of households | 846 | |
| Number of respondents | 846 | |
| Number of children below five | 213 | |
| Total population | 5431 | |
| *Respondent* | | |
| Age | 36.02 | 846 |
| % of respondents illiterate | 64.18 | 846 |
| *Toilet and Open Defecation* | | |
| % of households owing toilets | 5.32 | 45 |
| % of respondents reported open defecation | 95.15 | 805 |
| % of respondents reporting OD twice a day | 98.76 | 795 |
| *Hygiene Practices* | | |
| % of respondents reported handwashing with soap after defecation | 10.52 | 89 |
| % of households observed having a soap at the handwashing station | 39.48 | 334 |
| *Child Nutrition Indicators (for the panel of children aged 0-5 years)* | | |
| Age (months) | 23.15 | 213 |
| % girl child | 49.77 | 106 |
| % of children reported diarrhea | 26.76 | 57 |
| *Weight-for Age* | | |
| Mean | −1.45 | 213 |
| % of children with weight-for-age less than 1 standard deviation | 66.20 | 141 |
| % of children with weight-for-age less than 2 standard deviations (underweight) | 34.27 | 73 |
| % of children with weight-for-age less than 3 standard deviations | 14.08 | 30 |

Notes: Column (1) shows sample means of the responses obtained prior to any intervention in the experiment villages. One woman from each household was interviewed.

around 10% of the respondents reported washing their hands with soap after defecation. In terms of nutritional outcomes, around one-third of the children below five years of age in the program villages were underweight with a mean weight-for-age (WAZ) score being −1.45. 27% of the children reported having suffered from at least one diarrheal episode in the past 30 days.

Table 7 in Appendix 2 in S1 File reports the baseline averages for a host of household, respondent, and children characteristics across the treatment arms. Overall, the sample is balanced as treatment and control villages are quite similar.

## 4.2. Empirical strategy

The random assignment of villages to different treatment arms provides a straightforward causal interpretation of each intervention. We employ a difference-in-difference strategy of the following form where $y_{ihvt}$. is the outcome for individual $i$, residing in household $h$, village $v$ at time $t$:

$$y_{ihvt} = \beta_0 + \beta_1\ endline_t + \beta_2(treatment_A)_v + \beta_3(treatment_B)_v$$
$$+ \beta_4(treatment_A)_v \times endline_t + \beta_5(treatment_B)_v \times endline_t + \beta_6 X_{ihv}$$
$$+ \beta_7\ Gender\ Dummies + \gamma_i + \alpha_v + \epsilon_{ihvt}$$

Villages assigned to both BCC and toilet subsidy (cluster A) are captured through the dummy variable $treatment_A$ while $treatment_B$ indicates only toilet subsidy villages (cluster B). The coefficients $\beta_4$ and $\beta_5$ are the causal estimates of the intervention in both these clusters as compared to the control group. The key identification assumption here is that besides the two types of treatment, any time-varying factor will impact the relevant outcomes in the treatment and control groups in the same manner. Since $treatment_A = 1$ d $treatment_B = 1$ were randomly assigned, we expect $E\big(\epsilon_{ivt}|(treatment_A)_v \times endline_t\big) = 0$ and $E\big(\epsilon_{ivt}|(treatment_B)_v \times endline_t\big) = 0$. Under the parallel trends assution, we can test the significance of the treatment through the null hypothesis that $\beta_4 = 0$ and $\beta_5 = 0$.

We control for the vector of individual, household, and village-specific characteristics, $X_{ihv}$. In trying to estimate the impact of our intervention on child nutrition, we control for the mother's body mass index and education. Mother's literacy level accounts for her agency and awareness of best nutritional practices. At the household level, we control for the household head's age, education, religion, size of the landholding, and an index for durable assets. We also include gender dummies along with the child and village-level fixed effects ($\gamma_i$ and $\alpha_v$, respectively). Since the treatment was assigned at the village level, we cluster the standard errors by the village. We use a Wild-cluster bootstrapping procedure to bootstrap the standard errors on our coefficients of interest for improved inference with fewer clusters [44].

### 4.3. Outcomes

The study focuses on three primary outcomes (1) access to toilets, (2) open defecation, (3) hygiene practices, and their subsequent impact on one secondary outcome (4) child health outcomes.

1. To measure access to toilets in a village, we present results for the percentage of households in the village having access to a toilet.

2. Open defecation is measured through the percentage of respondents practicing OD or not using toilets. To ascertain this, we asked the respondents where they went for defecation in the previous 24 hours.

3. Our BCC intervention also influenced the hygiene behavior outcomes. Hygiene behavior was measured through the following outcomes: handwashing with soap (self-reported), presence of soap at the handwashing station, proper stool disposal, and frequency of using toilets (or going for OD).

   1. Responses for handwashing with soap were classified as a dummy: 1 for yes, and 0 for no.

   2. Similarly, the presence of soap at the handwashing station was captured through an indicator.

   3. Proper stool disposal (i.e., either flushing the child's feces in the toilet or burying it in the ground) was captured through a binary indicator 1 for proper stool disposal and 0 for improper stool disposal.

4. Finally, the frequency of going for urination or defecation was also included in the results as it baseline survey suggested that it varied from people practicing OD and the ones using toilets. People usually go to the field to urinate and/or defecate twice a day (either before sunrise or after sunset) in rural India. Access to toilets grants people the freedom to use them for urination or defecation at any time during the day. Hence, we constructed an indicator variable that captures if people went for urination or defecation at most twice a day (variable takes the value 0), and more than twice in one day (variable takes the value 1).

4. We measure the child's nutritional outcomes through changes in weight.

The weight of a child is measured using a weight-for-age z score (WAZ score), which compares the child's weight to the weight of healthy children of the same age and sex in a reference population, in terms of standard deviations. It is defined as

$$Z - Score = \frac{\text{Measured weight} - \text{Median weight for a reference age group}}{\text{Standard deviation of weight for reference age group}}$$

The WAZ score is useful to identify whether the child is underweight, normal weight, or overweight. A WAZ score of zero indicates that the child's weight is the same as the median weight of the reference group. A negative WAZ score means that the child's weight is lower than the median weight of the reference population. If a child's WAZ score is two standard deviations below that of the reference group, they are classified as underweight, which reflects a nutritional failure. To ensure accuracy, we follow the WHO's 2006 international reference population and exclude children whose WAZ scores are below minus six or above five standard deviations [45]. We further check for potential pathways of the increase in WAZ scores by using diarrhea prevalence for analysis.

## 5. Section 5: Results

We find that the BCC intervention was successfully disseminated in cluster A villages (see Appendix 1 in S1 File). Hence, we hypothesize that these households will not only *demand* greater *access* to toilets but also *adopt* them more successfully with the right set of hygiene practices.

### 5.1. Impact of treatments on ownership of individual household toilets and open defecation

Our findings report an increase in the percentage of households that built individual household toilets (from 2% to 98%) (or an increase from 13 to 332 toilets) (Fig 3a) and consequently led to a decline in respondents practicing OD in cluster A (from 98% to 4%) (or a decrease from 333 to 13 female respondents) (Fig 3b). There was also a near-universal decline in OD for the male respondents of the households at the endline in cluster A (see Fig 3, Appendix 3 in S1 File). All female respondents from cluster A villages who resided in households that had a toilet reported that all their family members used the toilets as well, in the endline.

In cluster B, the percentage of households with access to toilets improved by 81 percentage points (from 6% to 87%), but the percentage of respondents practicing OD fell only by 43 percentage points (from 95% to 52%).

In the control group, even though the individual household toilet access increased by 4.7%, the ODF rates remained about the same. This suggests that the increase in toilet adoption is not commensurate with the increase in toilet access, implying the need for hygiene promotion.

### 5.2. Impact of treatments on hygiene practices

Our BCC intervention is also hypothesized to have an impact on hygiene behavior outcomes. The DID estimates suggest there is a significant impact on hygiene practices. We find a 93% increase in self-reported handwashing with soap in cluster A after defecation relative to the control group (see Columns 1 and 2 of Table 3). The results for cluster B although significant are small in magnitude (relative to the control group).

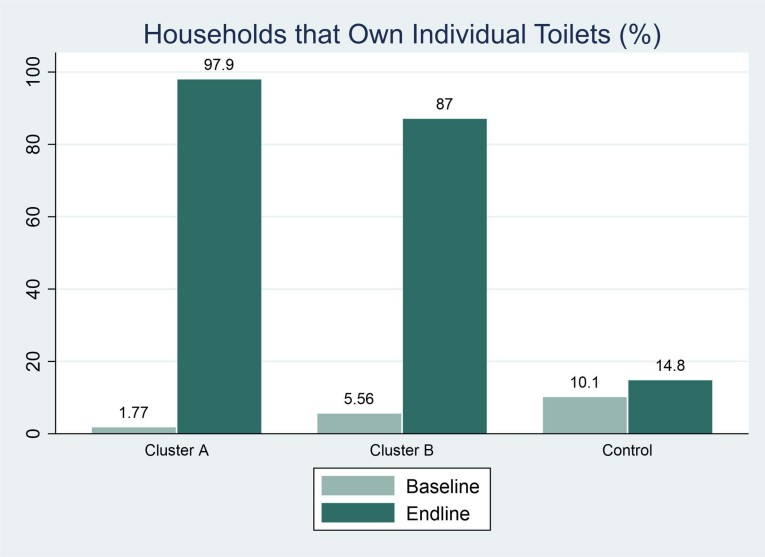

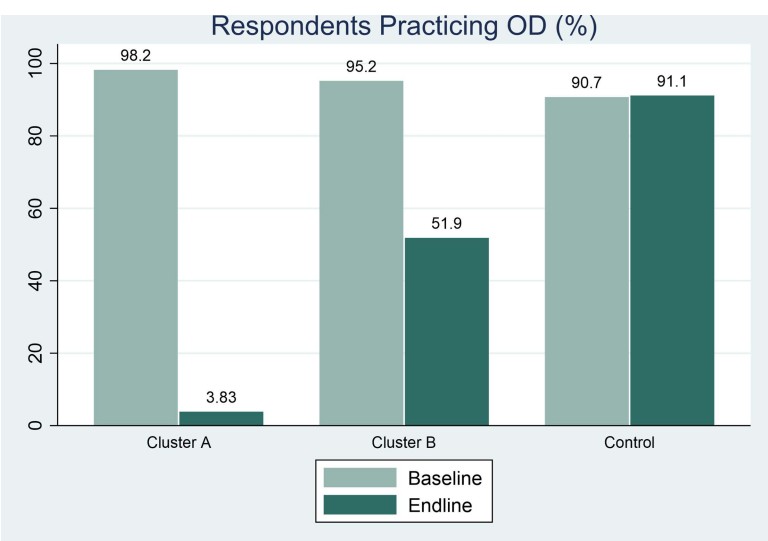

**Fig 3. (a) and (b): Impact of treatments on access to toilets and open defecation.**

Enumerator observation also corroborates this finding (although not to the same magnitude), as only 62% of households in cluster A had a soap present at their handwashing stations (see Columns 3 and 4 of Table 4). The difference could partly be explained as many respondents use washing detergents to wash hands after defecation. For our self-reported measure, we only counted handwashing with soap, and not detergent.

We also find a significant improvement in stool disposal as 83% of respondents in cluster A practice proper disposal techniques (i.e., either flushing in the toilet or burying it in the ground), relative to the control group. The results for cluster B are small and insignificant using Wild cluster bootstrapping (see Columns 5 and 6 of Table 3).

The last two columns of Table 3 estimate the results from an indicator variable that captures if the respondents practice urination or OD at most twice a day (variable takes the value 0), or more than twice in one day (variable takes the value 1). The estimates suggest that on

**Table 3.  Treatment impacts on hygiene behavior.**

| | Handwashing with soap after defecation (self-reported) | | Soap present at the handwashing station (enumerator observation) | | Proper stool disposal | | Practices urination and defecation more than twice a day | |
|---|---|---|---|---|---|---|---|---|
| | (1) | (2) | (3) | (4) | (5) | (6) | (7) | (8) |
| Importance of BCC + toilet construction (cluster A) | 0.93**** | 0.93**** | 0.62**** | 0.62**** | 0.83**** | 0.83**** | 0.91**** | 0.91**** |
| | (0.02) | (0.02) | (0.13) | (0.13) | (0.10) | (0.10) | (0.05) | (0.05) |
| *(Cluster-Robust p-Value)* | 0.000 | 0.000 | 0.000 | 0.000 | 0.000 | 0.000 | 0.000 | 0.000 |
| *(Wild Bootstrap p-Value)* | 0.000 | 0.002 | 0.001 | 0.0004 | 0.000 | 0.000 | 0.000 | 0.0004 |
| Importance of toilet construction (cluster B) | 0.03 | 0.03 | 0.08 | 0.08 | 0.19 | 0.19 | 0.43**** | 0.43**** |
| | (0.02) | (0.02) | (0.15) | (0.15) | (0.17) | (0.17) | (0.03) | (0.03) |
| *(Cluster-Robust p-Value)* | 0.13 | 0.14 | 0.58 | 0.58 | 0.21 | 0.21 | 0.000 | 0.000 |
| *(Wild Bootstrap p-Value)* | 0.07 | 0.07 | 0.56 | 0.54 | 0.15 | 0.15 | 0.001 | 0.001 |
| Individual fixed effect | Yes | Yes | | | Yes | Yes | Yes | Yes |
| Household fixed effect | | | Yes | Yes | | | | |
| Survey fixed effects | Yes | Yes | Yes | Yes | Yes | Yes | Yes | Yes |
| Village fixed effects | | Yes | | Yes | | Yes | | Yes |
| Unit of analysis | Respondent Level | Respondent Level | Household Level | Household Level | Child Level | Child Level | Respondent Level | Respondent Level |
| Observations | 1692 | 1692 | 1692 | 1692 | 426 | 426 | 1692 | 1692 |
| Villages | 15 | 15 | 15 | 15 | 15 | 15 | 15 | 15 |

*Notes*: Standard errors are reported in parenthesis below the coefficient and are clustered at the village-level.

\* p < 0.10,

\*\* p < 0.05,

\*\*\* p < 0.01,

\*\*\*\* p < 0.001.

average, 91% more respondents in Cluster A, and 43% more respondents in Cluster B practiced urination and OD more than twice in one day (relative to the control group).

## 5.3.  Impact on child weight

OD can lead to the contamination of water and food supplies with fecal matter [46]. To reduce OD, it is necessary to have access to and use improved sanitation facilities that prevent human waste from re-entering the environment. Studies have demonstrated that interventions aimed at preventing human waste from entering the environment can decrease the incidence of diarrheal diseases and other infectious diseases as well as child underweight or low weight-for-age z-scores [3,47,48].

We test for similar results in our study, by estimating the causal impact of the treatments on child WAZ scores using regression estimates. DID regression results in Table 4 exhibit the impact of the two treatments relative to the control group.

Column (1) is a simple difference-in-difference estimate. The coefficient of 0.69 suggests that as the community moves from universal OD to no OD, the WAZ score for children below significantly increases by 0.69 standard deviations for cluster A while remaining insignificant for cluster B. Both estimates are relative to the control group. Adding controls for the mother's BMI and education leads to a minor decrease in coefficient from 0.69 to 0.68 in column (2). As we progress from column (2) to (3) we add household controls, and the coefficients remain unchanged. Adding village fixed effects in column (4) also makes no change to the coefficient,

**Table 4. Impact of treatments on child weight-for-age z-scores.**

| | Child WAZ scores for children below five in both rounds | | | |
|---|---|---|---|---|
| | (1) | (2) | (3) | (4) |
| Importance of BCC + toilet construction (cluster A) | 0.69**/*** | 0.68** | 0.68** | 0.68** |
| | (0.29) | (0.30) | (0.30) | (0.30) |
| *(Cluster-Robust p-Value)* | 0.03 | 0.04 | 0.04 | 0.04 |
| *(Wild Bootstrap p-Value)* | 0.004 | 0.01 | 0.01 | 0.01 |
| Importance of toilet construction (cluster B) | 0.27 | 0.27 | 0.27 | 0.27 |
| | (0.24) | (0.24) | (0.24) | (0.24) |
| *(Cluster-Robust p-Value)* | 0.27 | 0.29 | 0.29 | 0.29 |
| *(Wild Bootstrap p-Value)* | 0.20 | 0.23 | 0.22 | 0.23 |
| Mother's BMI | | −0.03 | −0.03 | −0.03 |
| Mother's education | | 0.01 | 2.79 | 2.79 |
| Household Controls | | | Yes | Yes |
| Female child dummy | Yes | Yes | Yes | Yes |
| Child fixed effects | Yes | Yes | Yes | Yes |
| Survey fixed effects | Yes | Yes | Yes | Yes |
| Village fixed effects | | | | Yes |
| Observations | 426 | 426 | 426 | 426 |
| Villages | 15 | 15 | 15 | 15 |

Notes: Standard errors are reported in parenthesis below the coefficient and are clustered at the village-level.

\* p < 0.10,

\*\* p < 0.05,

\*\*\* p < 0.01

which is unsurprising as the villages received treatments randomly. All specifications include female child dummies and child-fixed effects. The level of significance for these coefficients remains the same for all the columns. The coefficients for cluster B remain insignificant for all the specifications. Therefore, on average, the WAZ scores increased by about 0.68-0.69 standard deviations for children below five in cluster A relative to the control group. Estimates for impact in cluster B are smaller and insignificant. The WAZ coefficient is not just statistically significant for cluster A but remains robust as we progressively include more controls and fixed effects.

Our estimated coefficient size of 0.68-0.69 increase in WAZ scores in cluster A echoes well with the magnitude of other studies that analyze the impact of the elimination of OD on child anthropometry outcomes. In Cambodia, the elimination of OD at the community level is associated with a 0.56 increase in WAZ scores for children below five [49]. Dickinson et al. 2015 report that India's national sanitation campaign, called the Total Sanitation Campaign, led to an increase in the ownership of toilets from 7% to 35% which led to a 0.26-0.31 increase in children's WAZ scores, after one year of implementation of the program. The lower magnitude of this coefficient could be due to lower increase in toilet ownership and also that their follow-up period was around six months in treatment villages, as opposed to almost one year in ours. Another study conducted in rural Mali with children below five years found that children who got the CLTS intervention had a 0.18 higher height-for-age z-score (HAZ) score than the control group, 1.5 years after the intervention [19]. Similarly, a study conducted in India found that the impact of the Total Sanitation Campaign (India's former sanitation campaign) led to an increase of 0.30 HAZ scores, 1.5 years after the intervention [7].

**5.3.1. Potential pathways.** Next, for the same sample of children, as we had in Table 4 (i.e. the panel of all children below the age of five in both baseline and endline), we evaluate possible biological channels linking improvement in the sanitary environment with better weight-for-age z-scores. When children are exposed to fecal contamination, they face a loss of net nutrition as calories are expended to fight infections (diarrhea, parasitic infections, or environmental enteropathy) [3,4]. This leads to loss of weight and hence, lower WAZ scores. If the elimination of OD is causing improvements in WAZ scores, then there should be a statistically significant improvement in intermediate outcomes like diarrhea.

Diarrhea: We replace the WAZ score with diarrhea in Column 4 of Table 4 and ran the difference-in-difference specification. Our findings report that the near-universal decline in OD in child's community is associated with a reduction of reported diarrhea of 28 percentage points (relative to the control group) at a 10% level of significance ($t = -1.86$; se = 0.15) while accounting for all potential confounders. Therefore, we believe that lower diarrhea was one of the potential mechanisms that led to higher WAZ scores in children.

## 6. Section 6: Discussion and policy implications

OD is a critical public health issue, contributing to adverse child health outcomes and its elimination has been enduringly challenging in India. Addressing this issue requires comprehensive approaches that address the credit and informational barriers faced by communities. In this study, we conducted a CRCT experiment with the objective of evaluating the effectiveness of removing both the credit and information constraints versus solely addressing the credit constraint at the universal level, eliminating OD, and improving child health outcomes in rural India. Our experiment contributes to the literature by empirically examining the precise nature of the intervention that can successfully eliminate OD in rural India.

The findings of our study indicate that removing the credit constraint alone had a limited decline in OD rates in cluster B. This result highlights the need to complement financial interventions with the provision of information and knowledge on proper sanitation practices. This is because OD is a deeply entrenched social norm in India and households may lack the information necessary to understand the health risks associated with OD and the benefits of building proper facilities. Hence, by removing both credit and information constraints simultaneously for *every* household, in cluster A led to the near-universal elimination of OD.

Next, we examine the impact of treatments on child health outcomes. The limited health impact reported in cluster B villages is consistent with experimental studies that have also found a limited impact of sanitation interventions on child health outcomes due to low adoption rates [10,17,20,22]. In cluster A, however, we estimated that as the community moves from no coverage to 100% coverage and adoption, it yields a 0.68-0.69 standard deviation increase in child weight-for-age z-scores. Our findings align with previous studies that emphasize the importance of information dissemination in sanitation interventions [8,43]. The result highlights the role of negative externalities associated with OD as children are fully protected from the adverse impact of OD only if the neighbors in their community also have access to and are adopting improved sanitation facilities [41]. This finding reiterates the importance of our research design which focussed on providing *universal* access to toilets as well as promoting its *universal* adoption.

The findings of our study have important policy implications. Addressing credit constraints alone, while beneficial, did not have as significant an impact as the combined intervention. This suggests that access to credit, without proper knowledge and awareness, may not be sufficient to drive sustained behavior change and eliminate OD in places where OD

is a social norm. The provision of information played a crucial role in instilling a long-term commitment to improved sanitation practices among the participants. Hence, to effectively eliminate OD, policymakers should prioritize the development and implementation of integrated approaches (i.e., financial support and information dissemination) that address multiple constraints simultaneously.

Policies related to the improvement of child health and nutritional outcomes in India have majorly considered these issues related to inadequate dietary intake. For instance, India has various national-level programs like, Integrated Child Development Services, National Health Mission, Poshan Abhiyan (translated as National Nutrition Mission), among others. These policies reflect the government's commitment to improving child health by addressing various aspects such as diversification of diets, adequacy of nutrient intake, immunization, and health-care access. These policies, although important, often overlook the complementary role of sanitation. So, even if the child is being fed an adequate diet, exposure to poor sanitation leads to infections and malabsorption of nutrients, leading to poor child health outcomes. This is exhibited by the heterogeneous impact of the interventions on child health in our experimental study. Our results strengthen the economic rationale of going beyond the traditional solely diet-based policies and investing in an improved sanitary environment in safeguarding the well-being and development of children.

Like most empirical studies, ours too has some limitations, and it is important to highlight that. The study does not have a 2 * 2 research design as our intervention did not include a BCC-only treatment. While including this treatment and evaluating its effectiveness would have been useful, we do not believe that it will change our main findings. Sanitation-related BCC campaigns without a concurrent subsidy have been proven to be an unsuccessful intervention in eliminating OD at the community-level [18,22]. Next, respondents' self-reported use of toilets might be subject to bias [50,51]. To mitigate this issue, we include child health outcomes in our analysis. Further research and scaled-up implementation of these interventions is recommended to consolidate the findings and improve sanitation outcomes across other developing nations that are also struggling with elimination of OD. Additionally, the long-term sustainability of such interventions could be further investigated through outcomes like height-for-age z-scores that take a longer time to improve.

In conclusion, this cluster-randomized controlled trial experiment demonstrates that removing both credit and information constraints is more effective than solely addressing the credit constraint in eliminating OD and bringing about subsequent gains in child health outcomes in rural India. Our findings align with previous research and provide valuable insights for policymakers and practitioners designing effective sanitation interventions in similar contexts.

## Supporting information

**S1 File. Appendix File.**
(DOCX)

## Acknowledgments

We thank the participants for their honest and enthusiastic participation in the study. We also thank Dr. Soumya Gupta, a research economist at Tata-Cornell Institute, Cornell University, and Dr. Andaleeb Rehman, a research associate at Tata-Cornell Institute, Cornell University for their insightful comments that greatly improved the manuscript. We also sincerely thank the reviewers for their insightful comments and suggestions, which have greatly enhanced the quality and clarity of this manuscript.

## Author contributions

**Conceptualization:** Payal Seth, Prabhu Pingali.

**Data curation:** Payal Seth.

**Formal analysis:** Payal Seth.

**Methodology:** Payal Seth.

**Supervision:** Prabhu Pingali.

**Writing – original draft:** Payal Seth.

**Writing – review & editing:** Prabhu Pingali.

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
