## [Decision Letter · Decision Letter 0]

29 Oct 2024

PONE-D-24-24792Making India Open Defecation Free by Alleviating Information and Credits Constraints in Toilet Access: Evidence from a Cluster Randomized Control TrialPLOS ONE

Dear Dr. Seth,

Thank you for submitting your manuscript to PLOS ONE. After careful consideration, we feel that it has merit but does not fully meet PLOS ONE’s publication criteria as it currently stands. Therefore, we invite you to submit a revised version of the manuscript that addresses the points raised during the review process.

ACADEMIC EDITOR:

Dear authors,

The authors have to revise in line with comments of the reviewers before re assessment.

With regards,

Ranjit

We look forward to receiving your revised manuscript.

Kind regards,

Ranjit Kumar Dehury

Academic Editor

PLOS ONE

Journal Requirements:

1. When submitting your revision, we need you to address these additional requirements. Please ensure that your manuscript meets PLOS ONE's style requirements, including those for file naming. The PLOS ONE style templates can be found at https://journals.plos.org/plosone/s/file?id=wjVg/PLOSOne_formatting_sample_main_body.pdf and https://journals.plos.org/plosone/s/file?id=ba62/PLOSOne_formatting_sample_title_authors_affiliations.pdf 2. Please include a complete copy of PLOS’ questionnaire on inclusivity in global research in your revised manuscript. Our policy for research in this area aims to improve transparency in the reporting of research performed outside of researchers’ own country or community. The policy applies to researchers who have travelled to a different country to conduct research, research with Indigenous populations or their lands, and research on cultural artefacts. The questionnaire can also be requested at the journal’s discretion for any other submissions, even if these conditions are not met.  Please find more information on the policy and a link to download a blank copy of the questionnaire here: https://journals.plos.org/plosone/s/best-practices-in-research-reporting. Please upload a completed version of your questionnaire as Supporting Information when you resubmit your manuscript. 3. Thank you for stating the following financial disclosure: “Bill and Melinda Gates Foundation (# OPP1137807).” Please state what role the funders took in the study.  If the funders had no role, please state: "The funders had no role in study design, data collection and analysis, decision to publish, or preparation of the manuscript." If this statement is not correct you must amend it as needed. Please include this amended Role of Funder statement in your cover letter; we will change the online submission form on your behalf. 4. In the online submission form, you indicated that [IThe data underlying the results presented in the study are available from Payal Seth at payalseth1309@gmail.com. or contact her at +91 9811391172.] All PLOS journals now require all data underlying the findings described in their manuscript to be freely available to other researchers, either 1. In a public repository, 2. Within the manuscript itself, or 3. Uploaded as supplementary information.This policy applies to all data except where public deposition would breach compliance with the protocol approved by your research ethics board. If your data cannot be made publicly available for ethical or legal reasons (e.g., public availability would compromise patient privacy), please explain your reasons on resubmission and your exemption request will be escalated for approval. 5. When completing the data availability statement of the submission form, you indicated that you will make your data available on acceptance. We strongly recommend all authors decide on a data sharing plan before acceptance, as the process can be lengthy and hold up publication timelines. Please note that, though access restrictions are acceptable now, your entire data will need to be made freely accessible if your manuscript is accepted for publication. This policy applies to all data except where public deposition would breach compliance with the protocol approved by your research ethics board. If you are unable to adhere to our open data policy, please kindly revise your statement to explain your reasoning and we will seek the editor's input on an exemption. Please be assured that, once you have provided your new statement, the assessment of your exemption will not hold up the peer review process. 6. Your ethics statement should only appear in the Methods section of your manuscript. If your ethics statement is written in any section besides the Methods, please move it to the Methods section and delete it from any other section. Please ensure that your ethics statement is included in your manuscript, as the ethics statement entered into the online submission form will not be published alongside your manuscript.

Additional Editor Comments:

Dear authors,

The authors have to revise in line with comments of the reviewers before re assessment.

With regards,

Ranjit

Reviewers' comments:

Reviewer's Responses to Questions

**Comments to the Author**

1. Is the manuscript technically sound, and do the data support the conclusions?

Reviewer #1: Partly

Reviewer #2: Yes

2. Has the statistical analysis been performed appropriately and rigorously? 

Reviewer #1: No

Reviewer #2: Yes

3. Have the authors made all data underlying the findings in their manuscript fully available?

Reviewer #1: No

Reviewer #2: No

4. Is the manuscript presented in an intelligible fashion and written in standard English?

Reviewer #1: No

Reviewer #2: Yes

5. Review Comments to the Author

Reviewer #1: Thank you for the opportunity to review this article. It is written on a very important topic and can potentially provide very important contributions to the field.

Unfortunately, I cannot recommend it for publication at this stage. Here are my suggestions to improve the manunscript:

1. The literature review has to be expanded. You are presenting current efforts to improve sanitation in India as only focused on the supply side of the problem. This is false. In fact, SBM had a big demand component, which is completely overlooked in the paper. When you refer to limited implementation of behavioral change, you need to be more specific about how your intervention is different from what was proposed/implemented before.

2. The paragraph on CLTS makes it seem like CLTS has never before been implemented in India. In fact, we know that many CLTS attempts failed (see for example Leong 2020, "Narratives of sanitation"). If your approach is merely CLTS, why is it different this time?

3. After reading the paper I am not confident I understand exactly what the intervention included. You need to add a section describing the intervention in more detail.

4. You chose a district with a 17% coverage rate. How is it that the coverage is so much lower in your sample (5.32%)?

5. How was the sample size determined? Why did you only choose 5 villages per cluster?

6. While I am not a great supporter of balance tests in studies, it is very important in a study with that small sample size. Why was the coverage in cluster A so much lower than in cluster B and control? 2% vs 10% is a massive difference. You further report sample sizes as 846 when coverage is a village-level variable so your sample can't be greater than 15. You need to motivate those differences in coverage.

7. I am very confused about your choice of a dependent variable. The title doesn't suggest anything about using health outcomes and makes it seem like you were testing every possible dependent variable to find any effect.

8. You mention monthly follow-ups. Why are they not included in the identification strategy? Why is only the endline included? What was the purpose of monthly follow-ups?

9. WAC is a very misleading variable. Is it plausible to fix malnutrition outcomes in 2 years? Children also aged in that period so you are comparing children at different ages in the baseline and endline. If OD disproportionately affects infants, you might see a correction due to aging /mortality. After 2 years of the study, how do you include new-born children who didn't have data at the baseline?

10. What was the mortality/attrition in the sample?

11. Have you considered desirability effects in responses? You mention yourself that observations didn't confirm hand-washing practices in a large number of cases but dismiss the finding. Is it possible that respondents over-report toilet use to make your surveyors happy?

The most important thing I want you to focus on is positioning your findings in the body of literature. CLTS has failed many times before in India. How is your intervention so successful while nothing else including SBM worked? If you indeed discovered a holy grail of OD in India, I have nothing but respect for you, but given the quality of the evidence I suspect your long-term efficacy will be modest at best.

If you indeed find your intervention to have an almost universal success, you need to make sure the recipe is publicly available and make the intervention details available to the readers.

Reviewer #2: 1. When using acronyms, please provide full form when it is used in the first place. For example, WAZ in the abstract.

2. There is confusion in the timeline-related description. On page no. 12, in the second paragraph authors have written “conducting a monthly survey… from February 2018 to December 2019.” And then, in the next paragraph, they wrote “The final endline survey was carried out in December 2018 and January 2019.” The confusion is why the endline survey was conducted in the middle of the monthly survey. And how these two types of surveys are different from each other?

3. Another confusion is you mentioned 846 households participated in the endline survey; then how exactly is the double number (1692) of households included in the analyses? Also, you reported 278 children below five years of age at one place while at the other place, you mentioned 426 children and again at a place, you wrote 213 children.

6. PLOS authors have the option to publish the peer review history of their article (what does this mean? ). If published, this will include your full peer review and any attached files.

**Do you want your identity to be public for this peer review?** For information about this choice, including consent withdrawal, please see our Privacy Policy .

Reviewer #1: No

Reviewer #2: **Yes: ** Imteyaz Ahmad

---

## [Author Response · Author response to Decision Letter 0]

13 Dec 2024

Reviewer #1: Thank you for the opportunity to review this article. It is written on a very important topic and can potentially provide very important contributions to the field.

Unfortunately, I cannot recommend it for publication at this stage. Here are my suggestions to improve the manunscript:

1. The literature review has to be expanded. You are presenting current efforts to improve sanitation in India as only focused on the supply side of the problem. This is false. In fact, SBM had a big demand component, which is completely overlooked in the paper. When you refer to limited implementation of behavioral change, you need to be more specific about how your intervention is different from what was proposed/implemented before.

Thank you for your feedback regarding the literature review. We understand that this is a gap which needed to be addressed properly. We have addressed this concern by adding a section that discusses all the barriers associated with the limited adoption of toilets in India, using the studies that did impact the evaluation of TSC and SBM campaigns. We also discuss the demand-side components of the SBM and how studies have documented that despite the success of SBM in enhancing access to toilets, its behaviour change either did not reach the beneficiaries or was coercive in nature. This is added from page 3 onwards of the document titled “Revised manuscript” or page 4 onwards of the document titled “Revised Manuscript with tracked changes”.

The same section discusses the contribution of our research design and how it builds on the limitations of the previous studies in detail. For instance, discussing points in brief- First, our intervention differentiates itself by specifically addressing the limitations observed in prior behavioral change strategies, such as the reliance on coercive methods or SBM reaching only limited beneficiaries (see page 6). Our research design focuses on the (1) rigorous implementation of all the activities of CLTS (as is shown in Appendix A) and (2) is disseminated to all the people in the villages. This is a more community-driven approach that focuses on intrinsic motivation and long-term sustainability.

Second, our intervention also differs from previously implemented programs. Unlike previous national sanitation campaigns that were heavily focussed on building toilets (Dickinson et al., 2015; Hammer & Spears, 2016; Patil et al., 2014) and found limited impact on adoption of toilets and child health improvement, our study includes a behavior change component. And unlike experiments in developing countries that implemented only CLTS-like behavior change programs and found that standalone CLTS intervention is not effective for toilet adoption (Cameron et al., 2019; Guiteras et al., 2015), we included both CLTS along with subsidised toilet construction. Hence our study is different from the previous studies as it is emphasising the joint efficacy of both universal behavior change (removal of information constraint) and building subsidised toilet construction (removal of credit constraint) for every household that is opting for the subsidy. The interventions, unlike previous studies, are provided to all the households in one of the intervention arms.

We have added an entire section on “Research Design and its Contribution” from page 6 of the document titled “Revised manuscript” or “Revised Manuscript with tracked changes”. This delineates all the contributions that our study is making when compared to previous interventions.

2. The paragraph on CLTS makes it seem like CLTS has never before been implemented in India. In fact, we know that many CLTS attempts failed (see for example Leong 2020, "Narratives of sanitation"). If your approach is merely CLTS, why is it different this time?

We acknowledge the reviewer’s observation regarding the mixed outcomes of previous Community-Led Total Sanitation (CLTS) interventions in India. Leong (2020) highlights that traditional CLTS methods, which rely heavily on invoking shame and disgust to motivate behavior change, often fail to produce sustainable results. This failure can be attributed to their coercive nature and a lack of cultural sensitivity, as well as their inability to address the deeply entrenched socio-cultural norms and environmental identities tied to sanitation practices.

Our approach builds on these lessons by addressing key limitations of earlier CLTS implementations:

1. Cultural Sensitivity and Positive Reinforcement:

Unlike traditional CLTS methods, which often alienate communities by focusing on negative emotions, our intervention emphasizes intrinsic motivation and positive messaging. We align sanitation practices with values such as dignity, health, and modernity, ensuring that behavior change resonates with local aspirations and identities. This is particularly important given the findings from Leong (2020), which demonstrate that cultural norms and environmental identities—such as preferences for open spaces (a “green identity”)—play a significant role in shaping sanitation behaviors.

You can see the details of the activities deployed under CLTS in this project and its successful dissemination in Table A1, Appendix 1. The table shows how nearly all the members of the community were present during all the activities of CLTS

Table A1. Intervention Check

Sample Mean Observations

(1) (2)

% of respondents participated in following BCC activities (in cluster A)

Fecal calculation 96.76 328

OD mapping 97.64 331

Transect walk 96.76 328

Calculating health expenses 97.94 332

ODF pledge/commitment 88.50 300

Handwashing training 96.46 327

Toilet maintenance training 88.20 299

Natural leader selection 74.34 252

Action plan 75.81 257

Fecal-oral demonstration 97.94 332

Follow-up monitoring 89.97 305

Arouse dignity (eating other's feces) 97.64 331

CLTS committee selection 58.70 199

2. Integration of Behavior Change with Infrastructure Support:

Previous CLTS campaigns often operated independently of toilet construction, leaving motivated households without the means to adopt improved sanitation practices. In contrast, our intervention combines universal subsidized individual household toilet construction with a community-led behavior change campaign. This integrated approach addresses both supply-side constraints (through subsidies) and demand-side barriers (through BCC), ensuring households have both the means and the motivation to adopt and sustain toilet use.

3. Universal Outreach and Inclusivity:

Traditional CLTS campaigns have often targeted specific groups within communities, leading to fragmented outcomes and limited behavioral diffusion. Our approach ensures universal coverage, engaging every household in the intervention clusters. This not only fosters collective accountability but also helps normalize toilet usage as a community-wide practice, reducing resistance from those who might otherwise continue open defecation. (See table A1, Appendix A1).

4. Focus on Sustainability:

As noted in Leong’s work, the effectiveness of shame-based tactics diminishes over time, and behavioral changes are often not "sticky." Our approach prioritizes long-term sustainability by fostering community ownership of sanitation outcomes and leveraging social norms to embed these changes into daily practices. This aligns with recommendations from the paper, which advocate bridging the gap between "green" and "technology" identities by framing toilet use as both environmentally and socially desirable.

In summary, our intervention represents an evolution of the CLTS framework, addressing the structural and cultural challenges that have hindered its success in the past. By integrating behavior change communication with universal access to subsidized toilets and focusing on positive, inclusive messaging, our approach seeks to achieve sustainable sanitation outcomes and foster an open defecation-free environment.

3. After reading the paper I am not confident I understand exactly what the intervention included. You need to add a section describing the intervention in more detail.

That is a great point. We have added an entire section on “Research Design and its Contribution” from page 6 of the document titled “Revised manuscript” or “Revised Manuscript with tracked changes”. We hope that this makes it easier for the readers to comprehend the mature of interventions. Besides adding the details of the interventions with more clarity, we have also added Table 1, which can make it easier for the readers to refer to the interventions.

4. You chose a district with a 17% coverage rate. How is it that the coverage is so much lower in your sample (5.32%)?

Thank you for your insightful observation. While the district has an overall coverage rate of 17%, it is important to note that this figure reflects a broad average across various town and villages in the district, some of which may have higher coverage rates than others. Given that our study specifically focuses on rural villages within the district, where access to resources and infrastructure may be more limited, it is expected that the coverage rate in these areas would be lower. Consequently, the sample from these rural villages reflects a lower toilet coverage of 5.32%, which is consistent with the challenges faced in these regions.

5. How was the sample size determined? Why did you only choose 5 villages per cluster?

The sample size was determined using a cluster-based sampling approach, where each cluster consisted of 5 villages. In each of these villages, we included every household within the village, with a total of 866 households for 15 villages. This approach ensured comprehensive coverage and a highly representative sample, as all households in each village were included, rather than just a random selection.

The decision to select 5 villages per cluster was primarily driven by budget constraints, which limited the number of villages we could incorporate while maintaining the statistical integrity of the study.

To ensure the chosen sample would still yield reliable results, we ran power calculations to assess whether the impact of the intervention could be detected at child health level, i.e. if the sample was powered enough to detect the impact on children’s WAZ scores. These calculations confirmed that the sample size, based on the full census of households, would be sufficient to observe the intended impacts.

6. While I am not a great supporter of balance tests in studies, it is very important in a study with that small sample size. Why was the coverage in cluster A so much lower than in cluster B and control? 2% vs 10% is a massive difference. You further report sample sizes as 846 when coverage is a village-level variable so your sample can't be greater than 15. You need to motivate those differences in coverage.

Thank you for highlighting the importance of balance tests, especially in studies with smaller sample sizes. We acknowledge your observation regarding the coverage differences between cluster A (2%) and control group (10%). While this difference may seem substantial at first glance, it is important to note that statistical analysis during the balancing test (table A2, Appendix 2) showed that this variation was not statistically significant, indicating that it is unlikely to have affected the overall findings of our study.

The figure of 866 (attrition of 20 households in endline = 846 households) represents the total number of households included in our study across the 15 villages, based on a complete census. That is, every household in all the villages received the interventions and are a part of baseline and endline surveys. So, the study sample size of 866 households is inclusive of all the households in the 15 villages. That is, no random sampling or selection of sampled households was done as the interventions and surveys occurred at the census level.

The coverage percentages we reported refer to the proportion of these households that had built individual household toilets, provides insight into the impact at the household level within each cluster. When we mention sanitation coverage with respect to 846 households, it is the same as coverage across 15 villages.

We have clarified this on page 7 of the document titled “Revised manuscript” or “Revised Manuscript with tracked changes”.

All 866 households across the 15 villages participated in the study, representing a comprehensive census of the entire population in these villages.

7. I am very confused about your choice of a dependent variable. The title doesn't suggest anything about using health outcomes and makes it seem like you were testing every possible dependent variable to find any effect.

Thank you for your feedback regarding the clarity of our choice of the dependent variable. Our primary objective was to examine the impact of alleviating both information and credit constraints on reducing OD and improving child health outcomes in rural India.

We understand that it did not come across like that and we have made minor changes in the abstract and introduction to give due importance to the child health outcomes.

Most importantly, we have added a subsection titled “Mechanism Linking Improved Sanitation and Child Health” that highlights the literature from biomedical sciences to understand the pathways through which water, sanitation, and hygiene interventions can lead to improved health. This will strengthen the rationale as to why our paper is also focussing on child health outcomes. You can find this section from page 5 of the document titled “Revised manuscript” and “Revised Manuscript with tracked changes”.

Taking your suggestion into account, we have revised the title to: Addressing Information and Credit Barriers to Making India Open Defecation Free and Improving Child Health: Evidence from a Cluster Randomized Trial in Rural India

8. You mention monthly follow-ups. Why are they not included in the identification strategy? Why is only the endline included? What was the purpose of monthly follow-ups?

We appreciate your question regarding the role of monthly follow-ups and their exclusion from the identification strategy.

The primary purpose of the monthly surveys conducted from February 2018 to December 2018 was to track trends in toilet adoption over time. These surveys focused only on collecting self-reported toilet usage data from household heads and primary female respondents, along with enumerator observations for verification. The aim was to monitor short-term variations in adoption behavior across interventions, genders, and seasons, providing a more granular understanding of behavioral shifts. However, these monthly follow-ups were not comprehensive surveys; they did not include the broader socioeconomic and hygiene-related modules that were part of the baseline and endline surveys. Detailed results from monthly surveys are discussed in Appendix A3.

We have clarified this on page 9 of the document titled “Revised manuscript” or page 10 of “Revised Manuscript with tracked changes”.

From February 2018 to December 2018, we conducted monthly surveys with households to track toilet adoption trends. These surveys involved interviewing both the household heads and the primary female respondents only regarding their toilet adoption. Additionally, enumerators verified toilet usage during these visits. This approach allowed us to capture variations in toilet adoption across different interventions, genders, and seasonal patterns, providing a comprehensive understanding of the trends over time. We did not ask about other modules from the baseline survey in these monthly surveys.

9. WAC is a very misleading variable. Is it plausible to fix malnutrition outcomes in 2 years? Children also aged in that period so you are comparing children at different ages in the baseline and endline. If OD disproportionately affects infants, you might see a correction due to aging /mortality. After 2 years of the study, how do you include new-born children who didn't have data at the baseline?

We understand your concern and while all the malnutrition outcomes like height-for-age cannot be fixed in the short period of 2 years, outcomes related to weight can be changed in a relatively sh

---

## [Decision Letter · Decision Letter 1]

12 Jan 2025

Addressing Information and Credit Barriers to Making India Open Defecation Free and Improving Child Health: Evidence from a Cluster Randomized Trial in Rural India

PONE-D-24-24792R1

Dear Dr. Seth,

We’re pleased to inform you that your manuscript has been judged scientifically suitable for publication and will be formally accepted for publication once it meets all outstanding technical requirements.

Kind regards,

Ranjit Kumar Dehury

Academic Editor

PLOS ONE

Additional Editor Comments (optional):

There is no need for submitting further report

Reviewers' comments:

Reviewer's Responses to Questions

**Comments to the Author**

1. If the authors have adequately addressed your comments raised in a previous round of review and you feel that this manuscript is now acceptable for publication, you may indicate that here to bypass the “Comments to the Author” section, enter your conflict of interest statement in the “Confidential to Editor” section, and submit your "Accept" recommendation.

Reviewer #2: All comments have been addressed

2. Is the manuscript technically sound, and do the data support the conclusions?

Reviewer #2: Yes

3. Has the statistical analysis been performed appropriately and rigorously? 

Reviewer #2: Yes

4. Have the authors made all data underlying the findings in their manuscript fully available?

Reviewer #2: Yes

5. Is the manuscript presented in an intelligible fashion and written in standard English?

Reviewer #2: Yes

6. Review Comments to the Author

Reviewer #2: All the concerns raised are addressed. Authors have provided point to point response to the queries of the reviewers.

7. PLOS authors have the option to publish the peer review history of their article (what does this mean? ). If published, this will include your full peer review and any attached files.

**Do you want your identity to be public for this peer review?** For information about this choice, including consent withdrawal, please see our Privacy Policy .

Reviewer #2: **Yes: ** Imteyaz Ahmad

---

## [Editor Report · Acceptance letter]

PONE-D-24-24792R1

PLOS ONE

Dear Dr. Seth,

I'm pleased to inform you that your manuscript has been deemed suitable for publication in PLOS ONE. Congratulations! Your manuscript is now being handed over to our production team.

Kind regards,

on behalf of

Dr. Ranjit Kumar Dehury

Academic Editor

PLOS ONE